# The Effect of a Short-Term Occupational Therapy Intervention on the Participation and Personal Factors of Preschoolers with Developmental Disabilities

**DOI:** 10.3390/children10081401

**Published:** 2023-08-17

**Authors:** Bosmat Soref, Gary L. Robinson, Orit Bart

**Affiliations:** 1Department of Occupational Therapy, School of Health Professions, Faculty of Medicine, Tel Aviv University, Ramat Aviv, Tel Aviv 69978, Israel; sorefbos@inter.net.il; 2The Child Developmental Unit, Clalit Health Services, Tel Aviv 6816323, Israel; robinson@zahav.net.il

**Keywords:** developmental disabilities, participation, preschoolers, occupational therapy intervention, intervention documentation, sensory–motor

## Abstract

Background: Preschoolers with developmental disabilities are referred to occupational therapy due to their decreased participation in daily life occupations. The purpose of this study was to evaluate the improvement in preschoolers’ participation and sensory-motor abilities following an occupational therapy intervention. Materials and Methods: A prospective cohort study of 38 preschoolers and their parents was conducted using an interrupted time-series design, including assessments at three time points: base line (upon referral to an occupational therapy assessment), pre-intervention, and post-intervention after 9–12 sessions of occupational therapy interventions. Children were evaluated with the Developmental Test of Visual–Motor Integration, as well as the balance and fine motor precision sub tests of the Bruininks–Oseretsky Test of Motor Proficiency. Parents completed the Children’s Participation Questionnaire and the Child Performance Skills Questionnaire. Each intervention session was documented by the therapists using the Documentation of Occupational Therapy Session Intervention form. Results: Significant improvement in children’s sensory–motor abilities were found in balance, visual integration, and fine motor precision post-intervention. There were also improvements in the measures of diversity, children’s independence, and parental satisfaction. Conclusions: A short-term occupational therapy intervention applied to preschoolers with developmental disabilities is effective in improving their sensory–motor abilities, as well as in promoting their participation in daily activities.

## 1. Introduction

Preschoolers with developmental disabilities are referred to occupational therapy services in the community due to the children’s decreased participation in everyday life occupations [1,2]. The term “developmental disabilities” [3] is used to describe children with various difficulties in performing activities that require fine or gross motor coordination and maintaining their own organization, as well as children with learning difficulties and difficulties with regulation or sensory processing. When these children are evaluated, they may be diagnosed with developmental coordination disorder (DCD), attention deficit hyperactivity disorder (ADHD), sensory modulation dysfunction (SMD), or learning disabilities (LDs) [4,5,6].

Participation is defined by the World Health Organization [7] as involvement in life situations and engagement in occupations of daily activities that are essential for development, life experience, and well-being. The concept of participation is multidimensional, referring to the typical fields of occupation, and includes objective dimensions such as the activities in which the person takes part, how often and how independent they are, as well as subjective measures of human experience such as interest and pleasure [8,9,10]. Studies have shown that children with developmental disabilities show a decrease in their participation in daily occupations compared to their peers with typical development [10,11,12]. Preschool children with DCD showed lower levels of independence and enjoyment, and their parents were less satisfied compared to typically developed peers [1,2]. In addition, children with LDs participated less frequently in school and home activities [13].

According to the Occupational Therapy Practice Framework (OTPF) [14], the main goal of occupational therapy interventions is to promote participation. Nevertheless, a review of the literature reveals that there is currently a paucity of evidence-based studies evaluating the effectiveness of occupational therapy interventions for children’s participation in daily activities [15,16,17]. Moreover, the few existing studies mostly examined children with severe impairments such as ASD (24%) and CP (28%), while only a minority of the studies examined children with developmental disabilities such as ADHD (6%) and DCD (7%) [15]. The majority of studies evaluated school-aged children [15,18] or focused on examining a certain therapeutic approach such as sensory integration (SI) [19,20] or cognitive intervention; for instance, cognitive orientation to daily occupational performance (CO–OP) [21,22].

The literature review shows that different models guide therapeutic interventions in occupational therapy. The therapeutic approaches are usually clustered in two main types: process-oriented and task-oriented. Process-oriented approaches (e.g., SI) are bottom-up and aimed at improving body function, whereas task-oriented approaches are top-down (e.g., Co-Op) and tend to focus on task performance. The research literature supports the advantage of the task-oriented approaches, or the combination of both approaches in promoting participation [23,24]. The research review also indicates that individually tailored interventions directed at promoting participation outcomes, rather than an indirect approach via a change in body functions, are essential [25,26].

The importance of promoting the child’s participation, as well as the lack of studies evaluating the effectiveness of interventions for promoting children’s participation, raise the need to further investigate the participation outcome measures of occupational therapy interventions, especially among preschool aged children. Therefore, the aim of this study is twofold: to evaluate the improvement in preschoolers’ participation and sensory–motor aspects as a result of a short-term occupational therapy intervention and to examine the contribution of the intervention’s foci to the children’s improvement in participation.

## 2. Materials and Methods

### 2.1. Study Design

A prospective cohort study with an interrupted time-series design was conducted, including assessments at three time points.

### 2.2. Participants

Thirty-eight preschoolers (aged 4.93 ± 0.43 years) who were referred to occupational therapy evaluation, and their parents, were recruited through a convenience sampling method. All children attended mainstream preschools and were referred to occupational therapy evaluation for developmental difficulties, including gross motor difficulties, clumsiness, visual motor difficulties, fine motor delays, sensory modulation dysfunction, attention deficit disorder, or learning disabilities. Children receiving emotional treatment interventions or physiotherapy, as well as children diagnosed with cerebral palsy, autism, blindness, or deafness, were excluded. Table 1 summarizes the participants’ demographic characteristics.

### 2.3. Measurements

#### 2.3.1. Documentation of Intervention

Documentation of Occupational Therapy Session Intervention (D.O.T.S.I.) [27]. The D.O.T.S.I. is a form used to document the intervention by the occupational therapist. For each session, the therapist fills out the form by first specifying the activity and completing the duration of the activity in minutes. It includes eight categories that refer to the occupational therapy domain and process: the treatment’s physical and social contexts; the intervention types (i.e., occupation-based activity, purposeful activity, preparatory methods, and consultation); client factors (i.e., neuromusculoskeletal and movement, sensory, and mental functions); performance skills (i.e., motor, process/cognitive, and communication); performance patterns (i.e., habits, routines, and roles); occupation areas (i.e., activities of daily living (ADL), instrumental ADL, education, leisure/play, social participation); and intervention strategies (i.e., create/promote, establish/restore, maintain, modify, and prevent). In this study, we summarized all session reports and calculated the percentage frequency of each item in the total intervention. The D.O.T.S.I. is a reliable and valid form for documenting occupational therapy interventions [28].

#### 2.3.2. Assessment of Sensory–Motor Personal Factors and Performance Skills

Developmental Test of Visual–Motor Integration (VMI) [29]. VMI is a graphic test to assess visual motor coordination. The test has three parts: copying, visual recognition, and motor accuracy. In the current study, only the first part was used. It is a developmental sequence of geometric forms to be copied with paper and pencil.

Bruininks–Oseretsky Test of Motor Proficiency (BOT2) [30]. The BOT2 is a test of gross and fine motor proficiency for both children and young adults within a range of 4–21 years of age. In this study, we used the sub-tests of Balance and Fine Motor Precision.

Child Performance Skills Questionnaire (PSQ) [31]. The PSQ is a questionnaire completed by parents of children aged 4–6 years. It comprises 34 items, divided into three skill domains: motor skills (10 items); processing skills (14 items); and communication skills (10 items). Using a Likert scale, parents are asked to rate how each item describes their child from 1 to 6, where a higher score indicates higher performance skills. In this study, we used the three total scores (motor, process, communication skills) that were calculated as the mean score of all the items in each domain. The PSQ has good internal reliability (Cronbach’s coefficient alpha 0.84–0.92) [31].

#### 2.3.3. Assessment of Participation

Children’s Participation Questionnaire (CPQ) [10]. The CPQ is a questionnaire completed by parents of children aged 4–6 years. It measures the child’s participation pattern in six occupation domains: Activities of Daily Living, Instrumental Activities of Daily Living, Play, Leisure, Social Participation, and Education. Five participation measures are yielded from the questionnaire: participation diversity, participation frequency, child independence, enjoyment, and parental satisfaction. In the current study, these measures were calculated as the mean total scores of all the 44 activities.

### 2.4. Procedure

The study was conducted at three child development centers. It was approved by the Institutional Review Board (IRB) of Clalit Health Services for Community Medicine (Project identification code 0203-15-com2) on 14 December, 2016, and by the IRB of Tel-Aviv University (approval date 26 January 2017) (clinical trial number NCT02774135). The present condition of waiting periods for evaluation and intervention enabled the researchers to conduct a prospective cohort study. The study examined the improvement in the child’s participation and sensory–motor aspects following the occupational therapy intervention, compared to a waiting period without intervention. Parents of children who were referred to occupational therapy evaluation were approached and asked to participate in this study while they were on the waiting list for evaluation. After receiving an explanation about the study, the parents signed a consent form. Each child was evaluated by the first author at a base line when referred to occupational therapy (Time 0) with the VMI, as well as the Balance and Fine Motor Precision sub tests of the BOT2. Pre-intervention (Time 1) and post-intervention (Time 2), the same evaluation was conducted by six occupational therapists who were trained on conducting the assessments. They were blinded to the assessment time and had no prior acquaintance with the participants. The inter-rater reliability among the six occupational therapists was assessed during the filmed evaluation of two participants; the agreement rates among the six occupational therapists were between 85% and 95%. In addition, at these three time points of the study, parents completed the CPQ and the PSQ.

The participating children received 9–12 weekly directed intervention sessions of 45 min each by trained occupational therapists. All sessions were performed individually in the clinic. The study was aimed at evaluating the efficacy of interventions performed in practice by therapists in the developmental centers. Therefore, the interventions were conducted in accordance with the typical practice used by occupational therapists in the developmental centers. The interventions were aimed at improving the child’s everyday activities and performance skills using varied therapeutic approaches according to the child’s specific needs and the therapist’s preferences. Intervention goals were set for all participants with respect to their therapeutic needs and the parents’ preferences. The interventions involved a combination of approaches: Ayers sensory integration, motor learning, and developmental and cognitive-behavioral, together with guidance or counseling for parents [32]. The interventions were client-oriented and aimed at encouraging the child’s active involvement and providing the child with the right challenge. Each intervention session was documented at the end of the session by the therapist who delivered the intervention using the D.O.T.S.I. The therapists who participated in the study were provided an explanation about the D.O.T.S.I. and instructions on how to fill the form. All documented interventions were summarized (see Table 2).

## 3. Data Analysis

A repeated measures ANOVA analysis with a Greenhouse–Geisser correction was conducted to evaluate changes in the outcome measures throughout the study periods. We calculated partial eta square, the values of which are typically referred to as small (0.01), medium (0.06), and large (0.14) [33]. Post-hoc analyses using the Bonferroni correction were conducted to compare the differences in the variables between the waiting period and the intervention period [34,35]. A linear regression was performed to examine the contribution of the intervention foci to predict improvements in the children’s participation in daily activities. For this purpose, a new variable was created by calculating the difference between the participation scores of each participant pre- and post-interventions. Since this study documented the practical work, we did not control for the length of time that elapsed between the two study times. As a result, the duration of the study intervals (the waiting time between Time 0–1 and the intervention time between Time 1–2) were not identical for all participants. To control for time effects, we compared the waiting time (*M* = 23.82 weeks, *SD* = 15.23) and the intervention time (*M* = 22.71 weeks, *SD* = 6.58). No significant difference was found between the times (*t*_(37)_ = 0.421 *p* = 0.676). The data were analyzed using IBM SPSS Statistics (Version 27); the significance level was set at α < 0.05.

## 4. Results

### 4.1. Documentation of the Interventions

On average, the participants received 11.50 (*SD* = 1.03) intervention sessions. All sessions were performed individually at the clinic. Almost 90% of the parents were present in the treatment room during the interventions. All participant data on the intervention’s goals were collected and clustered into categories according to the domain of occupation (e.g., ADL, Education, or Play) or the performance skills and client factors (e.g., motor) the intervention aimed to promote. The summary of the findings as presented in Figure 1 indicates that the treatment goal of promoting participation in education was set for 100% of the participants, and the goal of promoting participation in play/leisure was set for approximately 39% of the participants. In addition, the treatment goals for most of the children (over 95%) were aimed at promoting motor performance skills and clients’ motor ability factors. Using the D.O.T.S.I., the therapists documented each intervention session. We summarized all session reports and calculated the percentage frequency of each item in the total intervention (Table 2). The most frequent intervention type was purposeful activity. Most treatments included activities that involved neuromuscular skeletal functions and motor skills. The most frequent sub-category of occupation was leisure/play. The establish/restore strategy was the most-used intervention.

### 4.2. Sensory–Motor Personal Factors and Performance Skills

Repeated measures of ANOVA showed a significant difference between the means of the children’s sensory–motor abilities: balance, visual–motor integration, and fine motor precision (BOT2) at three time points (Time 0, 1, and 2). The large partial eta squares for the sensory–motor abilities indicate that 21–27% of the variance in scores is due to the time of study within the subject effect. Significance was also found in process skills but not in motor performance skills or communication skills (PSQ) (see Table 3). Post-hoc analyses using the Bonferroni correction revealed that, during the waiting period, there was no significant difference in any of the sensory–motor abilities. However, during the intervention period, there was a significant difference in balance (*p* = 0.027), visual–motor integration (VMI) (*p* < 0.001), and fine motor precision (*p* < 0.001) as assessed by the BOT2. However, in process skills, no significant difference was found in either of the two study periods as measured by the PSQ.

### 4.3. Participation

Repeated measures of ANOVA showed a significant difference between means for participation in diversity, children’s independence, and parental satisfaction at three time points (Time 0, 1, and 2). The medium-to-large partial eta squares for the participation’s measures indicate that 12–39% of the variance in scores is due to the time of study within the subject effect. No significance was found in the measures of participation frequency and child’s enjoyment (Table 3). Post-hoc analyses using the Bonferroni correction revealed that, during the intervention period, there was a significant difference in participation diversity (*p* = 0.002) and child independence (*p* = 0.008). Conversely, during the waiting period, there was no significant difference in any of the participation measures.

A linear regression was performed to examine the contribution of the intervention’s foci to the child’s participation in daily activities (Table 4). The intervention foci on personal factors and performance skills explained about 26% of variance in the improvement child’s independence and 26% of the variance in improvement parental satisfaction. R^2^ adjusted is typically interpreted as “the percent of variation in one variable explained by other variables” [36].

## 5. Discussion

The study was conducted in the community and ecologically documented the therapeutic practice conducted in the field. As part of the study, the participating children received an average of 11.5 individual treatments, but each received no less than nine sessions. In the current study, the number of treatments was found to be effective in promoting the child’s participation. This is in line with the literature, according to which in order for a therapeutic intervention to be effective, at least nine treatment sessions are required [17]. In the current study, most of the parents were present in the room during the sessions. The goals of the intervention were aimed at promoting the areas of occupation and were determined in collaboration between the parent and the child’s direct caregiver. In addition, the activities in the meetings were adapted to the goals set for the child and his/her abilities. The features of the intervention in the current study are consistent with evidence from the research literature that indicates these features as contributing to change following the therapeutic intervention. In the literature, several features of the intervention were found to contribute to the advancement and participation of the child. These features are not necessarily related to a particular method of intervention. Aspects found to be effective for promoting the child and his participation were, among others, cooperation with the parents, setting common goals aimed at promoting participation, and planning and personally adapting the intervention to the child’s needs and abilities [15,25]. The goals of the research, as determined in cooperation with the parents, were aimed at promoting learning and play. It was found that all the children in the study had a goal set in the field of education, and about 40% of them also had a goal set in the field of play. However, a small percentage of the parents chose goals of promoting ADL self-care activities or the social sphere. Since the goals were determined in collaboration with the parents, they represent the contents that occupy the parents of preschool children with moderate difficulties. It is possible that these treatment goals are related to the fact that the children were about to start school. The finding regarding goals in the field of education is supported by previous findings, according to which most parents of children with developmental difficulties express concern about the impact of motor difficulties on their child’s academic achievements and often choose goals of promoting learning [26,37]. Parents of school-age children with DCD describe many difficulties in dealing with daily routines at home [38,39]. The reason why the parents did not set goals to promote this domain may be related to the age of the children in the current study. A previous study showed that, in the area of ADL, the participation of kindergarten children with DCD does not differ from that of their peers with typical development [1]. Therefore, it is possible that, at a young age, parents do not experience the child’s need for support around self-care as unusual or worrisome and do not set a goal to promote participation in this domain of occupation.

The research findings indicate that the majority of the children were referred due to difficulties in their fine motor functions. The reason for referral can explain the treatment goals set for the children. Indeed, the goals of promoting motor skills and personal factors were set for all participating children. In accordance with these goals, the main focus of the treatment was directed at promoting motor personal factors and motor performance skills. There is evidence in the literature supporting the concept that children’s participation is related to difficulties in performance skills [38,40]. Hence the justification for directing resources in the intervention at their promotion. However, the evidence is not conclusive [41,42]. Yet, we find in the field that the therapists devote a significant part of the treatment to promoting personal factors and performance skills, using a remedial intervention strategy of establishing/improving [17,27]. This intervention strategy represents a therapeutic concept designed to change personal factors and performance skills in order to build or restore skills or abilities [8]. In the current study, it was found that the intervention strategy most used was establishment/improvement. Moreover, although the goals of the treatment are aimed at the areas of occupation, about 60% of the activities incorporated in the intervention were purposeful activity, followed by preparatory activity (about 20%), while only about 17% of the intervention incorporated an occupational activity. These findings are consistent with a previous study on the documentation of pediatric occupational therapy interventions [27,43]. The current research results indicate that a therapeutic strategy of establishment/improvement, which represents a bottom-up approach, is effective in improving children’s participation.

In the performance skills of motor, process, and communication, an increase in the mean score was measured throughout the two periods of the study. The research findings show that the improvement in process skills was significant. However, the change cannot be attributed to the intervention period. The finding regarding the lack of improvement in the child’s performance skills can be explained by the way the parents perceive the child’s skills. In a previous study, a disparity was found between the way the parents evaluate the child’s performance skills and that of the occupational therapists. The occupational therapists report higher performance skills than the parents. In addition, the study found that the report of the occupational therapists correlated with the skills measured in the clinic [43,44]. Therefore, it can be assumed that parents value children’s performance skills lower than their actual ability.

The children’s balance improved following the occupational therapy intervention. This finding reinforces previous findings showing that balance reactions improve following interventions aimed at practicing these skills [20,45,46,47]. In the current study, the intervention was not aimed primarily at promoting balance skills, but the treatment documentation indicates its focus. The research literature describes that practicing taekwondo or jumping on a trampoline promotes equilibrium reactions. The researchers explained that the jumps and turns practiced in these activities stimulate the balance system and improve its performance [45,48]. Similarly, the practice in the occupational therapy clinic allows the child to exercise a variety of activities such as jumping, spinning, walking on unstable surfaces, climbing ladders, etc., which improves their equilibrium responses. This way of working follows the intervention strategy of establishment/improvement, which guides the work of most clinics in the current study in particular and in child development in general [27].

In this study, following the intervention, an improvement was found in the children’s visuomotor integration and fine motor precision. Previous studies have shown that occupational therapy interventions improve visuomotor integration, as well as fine motor skills [20,49,50,51]. These findings are important since visuomotor integration and fine motor skills in preschool are important components of a child’s development and essential for school participation [51]. They are associated with activities such as drawing and writing and are among the factors that predict the child’s adaptation to school [29,52], hence the importance of promoting these abilities during the intervention with preschoolers. Indeed, in the current study, about 80% of the participation children were referred for occupational therapy intervention due to difficulties in fine motor skills, and all were assigned a treatment goal that included promotion of the education domain.

In this study, the child’s participation was examined following a short-term intervention in occupational therapy. The findings indicate an improvement in participation diversity, children’s independence, and parental satisfaction. In the participation measures of frequency of participation and the child’s enjoyment, an upward trend was found in the mean score, but no significant change was demonstrated. As mentioned, a review of the effectiveness of therapeutic interventions to promote participation reveals that few studies have tested and demonstrated effectiveness in treatment [15,25]. Researchers suggest that since the concept of participation is multidimensional, it can be measured in different ways. The present study presents evidence that occupational therapy treatment is effective in promoting participation. Using the CPQ questionnaire enables us to evaluate the improvement in various measures of participation [10].

The participation diversity expanded following the intervention. One of the explanations for this finding could be that the intervention targeted promoting participation goals. In the current study, a treatment goal of promoting participation in the domain of education was set for all participating children. This finding is consistent with the literature that emphasizes that intervention programs where the main outcome measure is the child’s participation have been proven to be effective for promoting participation [25]. It can be assumed that in accordance with the established goal, the intervention was aimed at promoting skills needed in the domain of education. Accordingly, it can be seen that the intervention was focused on motor and mental personal factors, as well as on motor and process performance skills. Despite this, interventions focusing on personal factors or on performance skills were unable to predict the improvement in participation diversity. This finding is consistent with the literature, according to which the main contribution to participation diversity is through environmental factors such as socioeconomic status and parental efficacy [12,44].

In the participation frequency, an upward trend was measured in the child’s mean score, but no significant improvement was demonstrated. Previous studies have shown that preschoolers with developmental difficulties did not differ from their typical development peers in the frequency of ADL and IADL [1,12,53]. This aspect could explain why no change in frequency was found. In accordance with the literature, in the present study, the mean scores for participation frequency at referral are within the normal range for preschoolers.

Children with developmental difficulties are described as less independent compared to their typical development peers [12,53,54]. Mothers of children with DCD describe their great need for support in order to encourage the child’s participation [38]. Since children with developmental difficulties need greater levels of assistance compared to their typical development peers, it can be assumed that an occupational therapy intervention aimed at promoting the child’s participation and improving his performance skills will improve his independence. As mentioned, the findings of the current study indicate that the child’s performance skills and personal factors improved following the intervention. Correspondingly, it was found that the child’s independence increased, requiring less assistance. An interesting finding is revealed when we examine the contribution of the intervention’s features to the child’s independence. When examining the focus of the intervention, a treatment focus on personal factors and performance skills was found to predict over 26% of the improvement in the child’s independence.

In the child’s enjoyment measure, an increasing trend was measured in the mean score, but no significant improvement was demonstrated. The mean scores for a child’s enjoyment at referral were found to be in the high range. Therefore, it can be assumed that the children had nowhere to improve.

Parental satisfaction with their child’s participation improved throughout the periods of the study, but with no difference between the waiting period and the intervention period. This finding is interesting because parents report an improvement in their satisfaction even before reporting an improvement in the objective measures of participation (i.e., diversity, frequency, and child’s independence). The literature review indicates that factors involving the child’s abilities, as well as environmental factors such as perceived parental efficacy, predict parental satisfaction with the child’s participation [12,53]. Correspondingly, parents of children with developmental difficulties report lower levels of satisfaction with the child’s participation compared to parents of children with typical development [53]. Parental satisfaction among parents of children with DCD was found to increase following treatment [17]. According to the current study, the improvement in parental satisfaction cannot be attributed to the period of the intervention. Therefore, it can be assumed that the change is related to the very beginning of the referral process for occupational therapy intervention. Since the satisfaction measure is a subjective one, it may have been affected by the fact that the child was referred and accepted for evaluation. The parent undergoing the procedure with the child experiences a sense of parental efficacy, which contributes to his satisfaction with the child’s participation. Nevertheless, it was found that the focus on personal factors and performance skills during treatment predicts almost 26% of the improvement in parental satisfaction. The focus on treating personal factors and performance skills may promote the child’s abilities and participation and thus contribute to the parents’ satisfaction with their child’s participation.

## 6. Conclusions and Limitations

### 6.1. Conclusions

The current study indicates that a short-term individualized occupational therapy intervention provided to preschoolers with developmental disabilities is effective in improving their balance, visuomotor integration, and fine motor precision. It is also effective in promoting their participation in daily activities. Participation diversity, children’s independence, and parental satisfaction were enhanced following the intervention. The findings indicate that interventions focused on children’s personal factors and performance skills contribute to their increased independence and parental satisfaction. Therefore, this study shows that OT interventions using a bottom-up strategy are effective in promoting children’s participation.

### 6.2. Limitations

The study included a sample of 38 participating children. Despite the small sample size, the research design of multiple measurements strengthens the results. Most of the participants belonged to families with a high or average socioeconomic background and lived in an urban area. This limits the generalizability of the research findings to other populations. Further studies may wish to design a Randomized Control Study with a larger sample size to assess the efficacy of occupational therapy intervention and its contribution to improved daily participation.

## Figures and Tables

**Figure 1 children-10-01401-f001:**
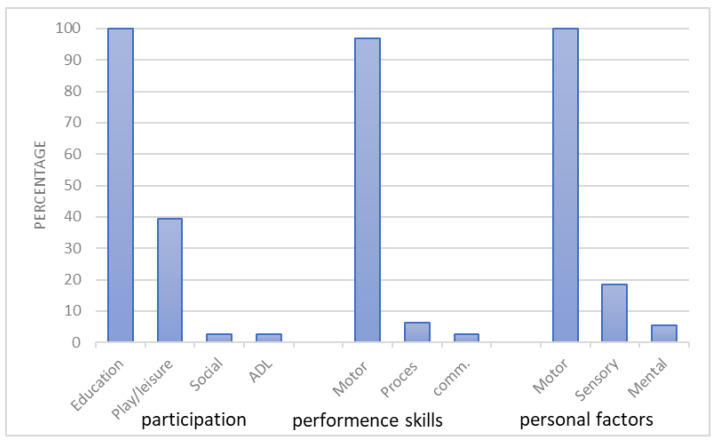
Prevalence of intervention goals to promote participation domains of occupations, performance skills, and personal factors.

**Table 1 children-10-01401-t001:** Participants’ demographic characteristics.

Characteristics (N = 38)		M (SD)	N (%)
Age	Child	4.93 (0.43)	
	Mother	35.66 (5.18)	
	Father	38.16 (5.13)	
Education (years)	Mother	13.94 (2.10)	
	Father	13.70 (2.48)	
Gender	Boy		29 (76.3)
	Girl		9 (23.7)
Marital status	Married		35 (92.1)
	Single parent		3 (7.9)
Religion	Jewish		37 (97.4)
	Christian		1 (2.6)
Religious	Non-observant		15 (39.5)
	Traditional		16 (42.1)
	Orthodox		5 (13.2)
	Ultra-Orthodox		2 (5.3)
Referral cause	Gross motor		2 (5.3)
	Fine Motor		31 (81.6)
	Emotional		2 (5.3)
	Attention		3 (7.9)

SD: Standard deviation.

**Table 2 children-10-01401-t002:** D.O.T.S.I. * Mean and standard deviation of each sub-category percentage for all sessions.

Category		Mean	SD
Intervention types	Occupation-based activity	16.81	19.22
	Purposeful activity	57.10	19.79
	Preparatory methods	20.43	14.78
	Consultation	5.66	6.05
Client factors	Neuromuscular skeletal/motor	51.13	14.57
	Sensory	14.92	9.15
	Mental function	33.95	15.41
Performance skills	Motor	55.15	14.46
	Process/Cognitive	36.21	10.62
	Communication	8.64	6.68
Areas of occupations	ADL/IADL	1.22	3.98
	Education	23.83	30.64
	Leisure/play	72.42	30.31
	Social participation	2.53	7.87
Intervention strategies	Create/promote	7.45	8.38
	Establish/restore	79.64	17.09
	Maintain	11.19	12.34
	Modify	1.47	3.43
	Prevent	0.25	0.87

* Documentation of occupational therapy session during interventions.

**Table 3 children-10-01401-t003:** Descriptive statistics of child’s personal factors, performance skills, and participation and repeated measures of ANOVA in the three time points of the study.

	Time 0 Mean (SD)	Time 1 Mean (SD)	Time 2 Mean (SD)	*p*	F (2,74)	Partial η^2^
BOT2						
Balance	10.39 (3.87)	11.37 (3.91)	13.68 (4.79)	<0.001	9.811	0.210
Visual–motor integration	93.11 (10.59)	91.61 (8.82)	97.53 (11.71)	<0.001	10.194	0.216
Fine motor precision	10.76 (4.10)	10.08 (3.25)	12.68 (3.60)	<0.001	13.667	0.270
Performance skills Questionnaire (PSQ)	4.97	5.08	5.03			
Motor	(0.58)	(0.63)	(0.71)	0.420	0.861	0.023
Processing	4.62 (0.61)	4.64 (0.86)	4.87 (0.78)	0.050	3.112	0.078
Communication	4.99 (0.66)	5.07 (0.91)	5.16 (0.73)	0.267	1.342	0.035
Participation (CPQ) Diversity	37.92 (2.63)	38.29 (2.14)	39.55 (1.74)	<0.001	13.23	0.263
Frequency	3.87 (0.29)	3.88 (0.38)	3.95 (0.37)	0.184	1.74	0.045
Child’s independence	5.12 (0.51)	5.21 (0.57)	5.39 (0.48)	<0.001	11.39	0.390
Child’s enjoyment	5.41 (0.33)	5.43 (0.43)	5.53 (0.48)	0.083	2.73	0.069
Parental satisfaction	5.22 (0.44)	5.36 (0.44)	5.44 (0.58)	0.012	4.93	0.118

BOT2: Bruininks–Oseretsky Test of Motor Proficiency.

**Table 4 children-10-01401-t004:** Model summary for explaining participation measures: diversity, independence, and parental satisfaction through intervention foci (n = 38).

	Predicting Variables	B	SE B	β	R^2^ _Adjusted_
Diversity	Motor factors	−0.092	0.170	−0.642	−0.104
	Sensory factors	−0.097	0.163	−0.423	
	Mental factors	−0.090	0.164	−0.663	
	Motor skills	0.120	0.091	0.815	
	Processing skills	0.087	0.101	0.444	
	Communication skills	0.106	0.094	0.353	
Child’s independence	Motor factors	−0.019	0.024	0.762	0.263 *
	Sensory factors	−0.026	0.023	−0.679	
	Mental factors	−0.025	0.023	−1.099	
	Motor skills	−0.034	0.013	−1.335 *	
	Processing skills	−0.026	0.014	0.792	
	Communication skills	−0.033	0.013	−0.637 *	
Parental satisfaction	Motor factors	0.088	0.026	3.260 **	0.257 *
	Sensory factors	0.076	0.025	1.760 **	
	Mental factors	0.083	0.025	3.220 **	
	Motor skills	−0.025	0.014	−0.897	
	Processing skills	−0.026	0.016	−0.692	
	Communication skills	−0.030	0.015	−0.532 *	

* *p* < 0.05, ** *p* < 0.01.

## Data Availability

Data will be available upon request.

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
