# Peer review of "The Effect of a Short-Term Occupational Therapy Intervention on the Participation and Personal Factors of Preschoolers with Developmental Disabilities"

_children, 2023, doi:10.3390/children10081401_

Round 1
Reviewer 1 Report
Dear authors,
An interesting article is presented for the scientific community and for society in general. The effect of a short-term occupational therapy intervention for children with mild disabilities is investigated.
I offer you some suggestions for improvement in some sections of your article:
- The title, despite being long, is clear with the purpose of the investigation.
- Keywords are enough. They allow the reader to quickly know the subject.
- The abstract should be 200 words. It exceeds 22 words. From my point of view, the summary is very well done and includes everything desirable: background, materials and methods, results and conclusions. I wouldn't take away words, but I leave this to the editor's decision.
- Throughout the document the authors must be included between numbered brackets []. This must be changed.
- The references used must be updated. Of a total of 43 cited articles, only 6 belong to the last 5 years (2018-2023). Despite how interesting the article was, it does not present current data. I strongly suggest you improve this by adding updated studies.
- The introduction adequately presents the theme and offers background, although these should be updated.
- The objectives of this research are viable and have been correctly developed.
- In table 1, the initials N, M and SD must be in italics. Also review and correct sections 3. Data analysis and 4. Results).
- Section 2, in general, is well done and meets the requirements of a research article.
- The tables and figures must have the same font as the rest of the document "Palatino Linotype". correct it.
- The discussion is presented in a clear and orderly manner. The findings link to previous studies, although as suggested above, add more current articles.
- The limitations of this study related to the sample are raised. Researchers are encouraged to include possible future research.
- The References section must be adapted to the MDPI regulations.

Author Response
We appreciate your comprehensive review and wish to thank you for your time and effort. Your comments further improved our manuscript.
An interesting article is presented for the scientific community and for society in general. The effect of a short-term occupational therapy intervention for children with mild disabilities is investigated.
I offer you some suggestions for improvement in some sections of your article:
- The title, despite being long, is clear with the purpose of the investigation.
Thank you.
In line with your comment and based on the second reviewer’s request we have shortened the manuscript title to: “The effect of occupational therapy intervention on participation and personal factors of preschoolers with developmental disabilities”.
- Keywords are enough. They allow the reader to quickly know the subject.
Thank you.
- The abstract should be 200 words. It exceeds 22 words. From my point of view, the summary is very well done and includes everything desirable: background, materials and methods, results and conclusions. I wouldn't take away words, but I leave this to the editor's decision.
To be on the safe side we have shortened the abstract. It includes now only 198 words.
- Throughout the document the authors must be included between numbered brackets []. This must be changed.
We have changed the reference style
- The references used must be updated. Of a total of 43 cited articles, only 6 belong to the last 5 years (2018-2023). Despite how interesting the article was, it does not present current data. I strongly suggest you improve this by adding updated studies.
We have updated the introduction and discussion and included more updated references.
- The introduction adequately presents the theme and offers background, although these should be updated.
We have updated the introduction
- The objectives of this research are viable and have been correctly developed.
Thank you.
- In table 1, the initials N, M and SD must be in italics. Also review and correct sections 3. Data analysis and 4. Results).
We have corrected it accordingly.
- Section 2, in general, is well done and meets the requirements of a research article.
Thank you.
- The tables and figures must have the same font as the rest of the document "Palatino Linotype". correct it.
We have changed it accordingly.
- The discussion is presented in a clear and orderly manner. The findings link to previous studies, although as suggested above, add more current articles.
We added more updated papers to the discussion section.
- The limitations of this study related to the sample are raised. Researchers are encouraged to include possible future research.
We have included suggestions for further studies to the conclusion section.
- The References section must be adapted to the MDPI regulations.
We have corrected the references section according to the MDPI regulations.

Reviewer 2 Report
I am very pleased to have the opportunity to review this manuscript. This manuscript is entitled "The effect of a short-term occupational therapy intervention on the participation, sensory-motor personal factors, and performance skills of preschoolers with mild disabilities. The manuscript, entitled "The effect of a short-term occupational therapy intervention on the participation, sensory-motor personal factors, and performance skills of preschoolers with mild disabilities," examines the effects of a short-term occupational therapy intervention.
I have read the manuscript and would like to comment as follows.
1.(Introduction)
The term "mild disabilities" sounds like it describes a degree of disability. Please consider adding that the degree of disability is not determined by the name of the diagnosis, but rather varies from person to person. In addition, please re-consider the title of the article.
2.(Terminology)
In the measurement section, a description of the evaluation is given from 2.3.1.-2.3.3. Abbreviations are used here, and it would be easier for the reader to understand if they were consistent with those used in the results.
3.(Result)
Since the process of the survey is explained in the methodology, please keep in mind that the order of presentation in the results section should follow the flow of the process. In other words, it is incongruous that D.O.T.S.I. is described at the beginning of the results.
4. (Statistics)
Please discuss the interpretation of the partial eta squares shown in Table 3, citing references as appropriate. Also, please discuss the contribution (R2 adjusted) shown in Table 4, citing previous studies, etc., as to how this 26% should be understood. If you have considered the possibility of multicollinearity(VIF) in your linear regression analysis, please add it. Also, please consider the interpretation of the results from the behavioral assessment of the subject child and the descriptive assessment of the parents, taking into account the bias that may affect the results. It is possible that parental satisfaction may have an impact on the PSQ and CPQ.
5. (misprint)
Please delete Lines 193-195 if they are not necessary. Please use the latest format (2023 edition) for your manuscript.
Author Response
We thank the reviewer for the supporting comments. We are sure that based on your comments the manuscript is improved.
I am very pleased to have the opportunity to review this manuscript. This manuscript is entitled "The effect of a short-term occupational therapy intervention on the participation, sensory-motor personal factors, and performance skills of preschoolers with mild disabilities. The manuscript, entitled "The effect of a short-term occupational therapy intervention on the participation, sensory-motor personal factors, and performance skills of preschoolers with mild disabilities," examines the effects of a short-term occupational therapy intervention.
I have read the manuscript and would like to comment as follows
1.(Introduction)
The term "mild disabilities" sounds like it describes a degree of disability. Please consider adding that the degree of disability is not determined by the name of the diagnosis, but rather varies from person to person. In addition, please re-consider the title of the article.
Based on your suggestion we have shortened the title to: “The effect of occupational therapy intervention on participation and personal factors of preschoolers with developmental disabilities”
We adopted your suggestion and we changed the term “mild disabilities” to “developmental disabilities”. In the sample section it is clearly described who participated in the study (children with Cerebral Palsy or Autism were excluded).
2.(Terminology)
In the measurement section, a description of the evaluation is given from 2.3.1.-2.3.3. Abbreviations are used here, and it would be easier for the reader to understand if they were consistent with those used in the results.
We have used the same evaluations’ abbreviation in the measures section, in the results section and in Tables, as suggested.
3.(Result)
Since the process of the survey is explained in the methodology, please keep in mind that the order of presentation in the results section should follow the flow of the process. In other words, it is incongruous that D.O.T.S.I. is described at the beginning of the results.
In oreder to adjust the process flow to the results flow, we moved the description of the D.O.T.S.I.to the beginning of the measures section.
- (Statistics)
Please discuss the interpretation of the partial eta squares shown in Table 3, citing references as appropriate.
We added to the data analyses section information on the partial eta square interpretation with reference. We also indicated its’ intrepertation in the result section
Also, please discuss the contribution (R2 adjusted) shown in Table 4, citing previous studies, etc., as to how this 26% should be understood. If you have considered the possibility of multicollinearity(VIF) in your linear regression analysis, please add it.
We added some information and reference to discuss the contribution of R2 adjusted to the result section.
We tested correlations among all the variables entered to the regression model and found that r was low to medium therefore we did not consider multiculiniarity. If neccary we can provide the table with the correlations among all predictors.
Also, please consider the interpretation of the results from the behavioral assessment of the subject child and the descriptive assessment of the parents, taking into account the bias that may affect the results. It is possible that parental satisfaction may have an impact on the PSQ and CPQ.
We are not sure we understand this comment. However, the main explanatory variables of Parental satisfaction were motor factors, sensory factors and mental factors. These factors were assessed by experienced therapists using standard measures. Only the PSQ (predictor variable) was completed by parents and its contribution to the explained variance is limited.
- (misprint)
Please delete Lines 193-195 if they are not necessary.
We have deleted these lines as suggested
Please use the latest format (2023 edition) for your manuscript.
We edited our manuscript using the latest format (2023 edition).
